# *ANO4* Expression Is a Potential Prognostic Biomarker in Non-Metastasized Clear Cell Renal Cell Carcinoma

**DOI:** 10.3390/jpm13020295

**Published:** 2023-02-07

**Authors:** Ahmed H. Al Sharie, Yazan O. Al Zu’bi, Tamam El-Elimat, Kinda Al-Kammash, Alma Abu Lil, Israa H. Isawi, Sarah Al Sharie, Balqis M. Abu Mousa, Abubaker A. Al Malkawi, Feras Q. Alali

**Affiliations:** 1Faculty of Medicine, Jordan University of Science and Technology, Irbid 22110, Jordan; 2Faculty of Pharmacy, Jordan University of Science and Technology, Irbid 22110, Jordan; 3Faculty of Medicine, Yarmouk University, Irbid 21163, Jordan; 4College of Pharmacy, QU Health, Qatar University, Doha 2713, Qatar

**Keywords:** anoctamin 4, transmembrane proteins with 16 domains, clear cell renal cell carcinoma, The Cancer Genome Atlas Program

## Abstract

*Background*: Over the past decade, transcriptome profiling has elucidated many pivotal pathways involved in oncogenesis. However, a detailed comprehensive map of tumorigenesis remains an enigma to solve. Propelled research has been devoted to investigating the molecular drivers of clear cell renal cell carcinoma (ccRCC). To add another piece to the puzzle, we evaluated the role of anoctamin 4 (*ANO4*) expression as a potential prognostic biomarker in non-metastasized ccRCC. *Methods*: A total of 422 ccRCC patients with the corresponding *ANO4* expression and clinicopathological data were obtained from The Cancer Genome Atlas Program (TCGA). Differential expression across several clinicopathological variables was performed. The Kaplan–Meier method was used to assess the impact of *ANO4* expression on the overall survival (OS), progression-free interval (PFI), disease-free interval (DFI), and disease-specific survival (DSS). Univariate and multivariate Cox logistic regression analyses were conducted to identify independent factors modulating the aforementioned outcomes. Gene set enrichment analysis (GSEA) was used to discern a set of molecular mechanisms involved in the prognostic signature. Tumor immune microenvironment was estimated using xCell. *Results*: *ANO4* expression was upregulated in tumor samples compared to normal kidney tissue. Albeit the latter finding, low *ANO4* expression is associated with advanced clinicopathological variables such as tumor grade, stage, and pT. In addition, low *ANO4* expression is linked to shorter OS, PFI, and DSS. Multivariate Cox logistic regression analysis identified *ANO4* expression as an independent prognostic variable in OS (HR: 1.686, 95% CI: 1.120–2.540, *p* = 0.012), PFI (HR: 1.727, 95% CI: 1.103–2.704, *p* = 0.017), and DSS (HR: 2.688, 95% CI: 1.465–4.934, *p* = 0.001). GSEA identified the following pathways to be enriched within the low *ANO4* expression group: epithelial–mesenchymal transition, G2-M checkpoint, E2F targets, estrogen response, apical junction, glycolysis, hypoxia, coagulation, KRAS, complement, p53, myogenesis, and TNF-*α* signaling via NF-*κ*B pathways. *ANO4* expression correlates significantly with monocyte (*ρ* = −0.1429, *p* = 0.0033) and mast cell (*ρ* = 0.1598, *p* = 0.001) infiltration. *Conclusions:* In the presented work, low *ANO4* expression is portrayed as a potential poor prognostic factor in non-metastasized ccRCC. Further experimental studies should be directed to shed new light on the exact molecular mechanisms involved.

## 1. Introduction

Renal cell carcinoma (RCC) is the most common genitourinary tumor in the United States, with a projected 79,000 new cases and 13,920 deaths in the year 2022 [1]. There is a twofold increase in the likelihood of men being diagnosed with kidney cancer as compared to women [1]. About 60% of cases are detected incidentally and 20% to 30% of patients present with metastatic disease [2]. In most cases, metastatic renal cancer is incurable [2,3]. Rather than being a single disease, RCC refers to a group of heterogeneous tumors that arise from the renal epithelium [4,5]. Each tumor subtype exhibits a unique histology, clinical course, and genetic profile, as well as a different therapeutic response [4,5,6]. Three major histological subtypes of RCC were identified, namely clear cell RCC or kidney renal clear cell carcinoma (ccRCC or KIRC), papillary RCC or kidney renal papillary cell carcinoma (pRCC or KIRP), and chromophobe RCC or kidney chromophobe (chrRCC or KICH) [7,8,9]. The most prevalent and aggressive subtype is the ccRCC, which accounts for more than 70% of all cases [4]. In addition, sarcomatoid and rhabdoid RCC tumors are associated with poor prognosis and resistance to targeted therapies [10].

ccRCC is one of the most inscrutable types of cancer [11]. An array of genetic or acquired factors can contribute to its development [11]. In terms of acquired risk factors, smoking, hypertension, obesity, chronic analgesic use, and diabetes are the most prevalent [11]. Genetically, ccRCC tumors exhibit significant mutation heterogeneity [12]. The vast majority of patients with ccRCC have a deletion in the short arm of chromosome 3 (loss of 3p) [13]. The two most common genetic abnormalities implicated in ccRCC are *VHL* (von Hippel–Lindau tumor suppressor) and *PBRM1* (protein polybromo1) [11,12]. Other genomic alterations involve *SETD2* (SET domain containing 2, histone lysine methyltransferase), *KDM5C* (lysine demethylase 5C), or *BAP1* (BRCA1 associated protein 1) [14,15]. The primary treatment for localized ccRCC is surgical resection with no role for conventional chemotherapy and radiotherapy [2,16,17].

Despite the development of numerous drugs targeting genomically prioritized pathways, patients with ccRCC have had limited responses to these treatments [2,18]. These results demonstrate that ccRCC tumorigenesis is a complex process, and hence, a thorough genomic, epigenomic, transcriptomic, and proteogenomic analysis is still further required to fully understand this cancer type in order to discover a curative treatment [13,19]. Therefore, the objective of this study is to evaluate the role of anoctamin 4 (*ANO4*) expression as a potential prognostic biomarker in non-metastasized ccRCC. *ANO4* belongs to the anoctamin (anion channels with 8 transmembrane domains) family [20]. It was formerly known as *TMEM16* (transmembrane proteins with 16 domains) [20]. In vertebrates, it includes ten paralogues with high sequence conservation [21]. Despite their close relationship, these proteins perform different functions and show distinct tissue distributions [22]. For example, some members such as *ANO1* and *ANO2* act as Ca^2+^-dependent ion channels, whereas others including *ANO6* function as Ca^2+^-dependent scramblases [23]. The roles played by other family members remain unclear and controversial [21]. However, several diseases have been associated with anoctamin proteins, including cancer, muscular dystrophy, asthma, arthritis, and epilepsy [20,24]. *ANO4* is primarily expressed in the central nervous system and certain endocrine glands, and it has been linked to a variety of neurological disorders [21]. To date, no study has evaluated *ANO4* expression as a potential prognostic biomarker in ccRCC.

## 2. Materials and Methods

### 2.1. Clinical and Transcriptomic Data Acquisition and Processing

This work aims to investigate the prognostic utility of *ANO4* mRNA expression in non-metastasized ccRCC (KIRC). KIRC clinical and transcriptomic data from The Cancer Genome Atlas Program (TCGA) were accessed using the California Santa Cruz Cancer Genomics Browser (UCSC Xena, http://xena.ucsc.edu, accessed on the 1 October 2022); a web-based platform for visualizing and analyzing public genomic data resources [25]. Experimental genotypic profiling was performed using Illumina HiSeq 2000 RNA sequencing platform to obtain level 3 data. *ANO4* expression from tumor samples with normal adjacent tissues was downloaded in an RNA-Seq by expectation maximization (RSEM) normalized count transformed as $\log_{2}\;(x+1)$. Clinicopathological variables incluedg age, gender, American Joint Committee on Cancer (AJCC) stage, International Society of Urologic Pathologists (ISUP) grade alongside the TNM scoring system which comprises tumor size, lymph node involvement, and metastasis status. Primary end points were the overall survival (OS), progression-free interval (PFI), disease-free interval (DFI), and disease-specific survival (DSS). The KIRC cohort was curated by omitting patients with an OS time of 0, metastasis status of M1 and MX, and patients without *ANO4* expression data. KIRC cohort was also divided into two subsets based on *ANO4* expression (high vs low expression), with a cut-point determined by X-tile software [26].

### 2.2. Statistical Analysis

IBM SPSS statistical package for Windows v.26 (Armonk, NY, USA) and GraphPad prism v.9.3.1 (San Diego, CA, USA) were utilized for statistical analysis and graph generation. Nominal data were presented as frequency (percentage). Mean ± standard deviation of the mean (SD) or standard error of the mean (SEM) were used to present normally distributed continuous variables, while non-normally distributed data were presented as median (interquartile range (IQR)). Normality was evaluated using the Kolmogorov–Smirnov test and the Shapiro–Wilk test aided with quantile–quantile (Q–Q) plots. Comparison between *ANO4* expression status against clinicopathological variables was performed as follows: *Chi*-square test or Fisher’s exact test for categorical variables, paired *t*-test for normally distributed paired samples, unpaired *t*-test and Welch’s corrected unpaired *t*-test for normally distributed non-paired samples according to variance equality, and finally, Wilcoxon matched pairs test and Mann–Whitney *U*-test for non-normally distributed data.

Kaplan–Meier survival methods were utilized to evaluate the impact of *ANO4* expression status in relation to OS, PFI, DFI, and DSS. Statistical difference across survival curves was detected using a log-rank test reporting the *p*-value, 95% confidence intervals (95% CI), and hazard ratios (HR). Univariate and multivariate Cox logistic regression models were applied to test clinicopathological variables and *ANO4* expression as independent prognostic indicators. Variables were dichotomized for Cox logistic regression analysis as follows: age (ref. ≤ 53 years), gender (ref. Female), pT stage (ref. T1 + T2), pN stage (ref. N0), AJCC stage (ref. Stage 1 + 2), ISUP grade (ref. Grade 1 + 2), and *ANO4* expression (ref. Low). The age cut-off value (53 years) was defined previously based on age-related differentially expressed genes optimized by the TCGA and Surveillance Epidemiology and End Results (SEER) database [27]. All statistical tests were two-sided and a *p* ≤ 0.05 was considered statistically significant.

### 2.3. Gene Set Enrichment Analysis (GSEA)

GSEA was performed to discern a set of molecular mechanisms involved in the prognostic *ANO4* signature. GSEA performs genome-wide transcriptional profiling across two expressional groups (high vs low *ANO4* expression) against a set of genes representing pivotal processes involved in oncogenesis/tumorigenesis [28]. Three molecular signatures databases were involved in the analysis, namely: hallmark gene sets, the C2 positional gene sets (Kyoto Encyclopedia of Genes and Genomes (KEGG) pathways in cancer), and C5 ontology gene sets (BP (biological process), CC (cellular component), and MF (molecular function)). The number of permutations was set to 1000 with the “gene set” permutation type. The chip platform was Human_UniProt_IDs. Gene sets were considered significantly enriched with an adjusted *p* value ≤ 0.05 and a false discovery rate (FDR) < 0.25.

### 2.4. Protein–Protein Interaction (PPI) Network Construction

Identification and retrieval of *ANO4*-related interacting genes were browsed using STRING v.11 (https://string-db.org, accessed on 1 October 2022), an online PPI networking resource based on functional interactions annotated using gene-enrichment analysis, GO/KEGG classification systems, and high-throughput text-mining [29]. Significant interactions were labeled if a combined score ≥ 0.4 was observed. The maximum number of interactions was limited to 50.

### 2.5. Immune Infiltration Analysis

Immune cell infiltration was enumerated from transcriptomes using xCell, an online platform with pre-calculated TCGA immune infiltration estimates that deploys a curve-fitting approach with a novel spillover compensation method [30]. Spearman’s correlation test was used to report Spearman’s rank correlation coefficient (rho, *ρ*) and the correlation significance.

## 3. Results

The presented work studies the usefulness of *ANO4* expression as a potential prognostic biomarker in non-metastasized ccRCC using a modified TCGA-KIRC cohort. It consisted of 422 patients with a median age of 61 (IQR: 51–71) and a male predominance (275, 65.2%). The majority of patients with non-metastasized ccRCC were found to have tumor grade 2 (204, 48.3%), followed by grade 3 (162, 38.4%) and grade 4 (39, 9.2%). In regard to tumor staging, the cohort was mainly distributed across two stages including Stage 1 (243, 57.6%) and stage 3 (121, 28.7%), which was definitely in concordance with the pathological T and N scoring distribution. Table 1 represents the baseline demographical and clinicopathological characteristics of the modified TCGA-KIRC cohort. X-tile software divided the patients based on *ANO4* expression harboring two groups (low expression, *n* = 126 and high expression, *n* =296).

### 3.1. Low ANO4 Expression Is Correlated with Poor Clinicopathological Features in Non-Metastasized ccRCC

In comparison with normal kidney tissue, the expression of *ANO4* was significantly upregulated (*p* < 0.0001) in the non-metastasized KIRC (Figure 1A). The latter finding was confirmed by analyzing the expression of *ANO4* using age and gender-matched participants (*p* < 0.0001), as illustrated in Figure 1B. Although tumor tissues exhibit a high *ANO4* expression compared to normal tissue, intriguingly, low *ANO4* expression within tumor samples is associated with advanced demographics and clinicopathological features. *ANO4* expression did not exhibit a tractable significant difference across age subgroups (*p* = 0.1989), as seen in Figure 1C. On the other hand, samples obtained from male participants have a significant low *ANO4* expression (*p* < 0.0001) in contrast to female samples (Figure 1D). The advanced tumor grade group, including grades 3 and 4, has significant reduced *ANO4* expression (*p* = 0.0497) in comparison to grades 1 and 2 (Figure 1E). The same pattern tends to be observed also in the tumor stage (*p* = 0.004, Figure 1F) and pT (*p* < 0.0048, Figure 1G). Despite the previous associations, a non-significant difference across groups with and without lymph involvement (*p* = 0.524) was observed (Figure 1H). Table 2 summarizes the differences between low and high-expression groups based on the dichotomized demographics and clinicopathological features.

### 3.2. Low ANO4 Expression Is Associated with Poor OS, PFI, and DSS

Kaplan–Meier survival analysis was used to explore the impact of *ANO4* expression (low vs high expression) on the OS, PFI, DFI, and DSS. The low *ANO4* expression group has a poor OS (HR = 2.093, 95% CI: 1.368–3.203, *p* = 0.0001) compared to the high expression group (Figure 1I). Of note, the same finding was observed regarding the PFI (HR = 2.485, 95% CI: 1.544–4.001, *p* < 0.0001, Figure 1J) and the DSS (HR = 3.879, 95% CI: 2.068–7.278, *p* < 0.0001, Figure 1K). However, the DFI did not remarkably differ across test groups (HR = 1.422, 95% CI: 0.4486–4.505, *p* = 0.513) as seen in Figure 1L. Further analysis involving curves stratification based on the dichotomized demographics and clinicopathological features was performed. Such analysis focuses on the effect of *ANO4* expression on the OS in specific sub-groups. In patients below the age of 53, *ANO4* expression did not impact the OS (HR = 1.715, 95% CI: 0.5709–5.153, *p* = 0.268, Figure 2A). On the contrary, the low *ANO4* expression group has a poor OS in the subgroup with an age above 53 (HR = 1.977, 95% CI: 1.257–3.109, *p* = 0.001, Figure 2B). Furthermore, gender-related OS analysis revealed a significant difference among the expression groups with the same effect of low *ANO4* expressing (Male: HR = 1.758, 95% CI: 1.055–2.931, *p* = 0.0203) vs (Female: HR = 3.246, 95% CI: 1.421–7.416, *p* < 0.0001) as presented in Figure 2C,D, respectively. In both low- and high-grade groups, low *ANO4* expression still exhibits the same survival effect (G1 + G2: HR = 1.946, 95% CI: 0.9375–4.038, *p* = 0.0378) vs (G3 + G4: HR = 1.972, 95% CI: 1.180–3.296, *p* = 0.0043) as presented in Figure 2E,F, respectively. While the low-stage group follows the same low *ANO4* expression effect (Stage 1 + 2: HR = 2.025, 95% CI: 1.086–3.775, *p* = 0.0085, Figure 2G), a non-significant survival difference was observed in the high-stage group (Stage 3 + 4: HR = 1.712, 95% CI: 0.9721–3.016, *p* = 0.0518, Figure 2H). The same effect was also in harmony regarding the pT, that low *ANO4* expression is still a poor survival indicator in both low pT (T1 + T2: HR = 2.035, 95% CI: 1.105–3.747, *p* = 0.0072, Figure 2I) and high pT (T3 + T4: HR = 1.752, 95% CI: 0.9844–3.117, *p* = 0.0444, Figure 2J). Finally, survival curves generated in sub-sets with and without lymph node involvement were impacted in the same trend by low *ANO4* expression (N0: HR = 2.045, 95% CI: 1.155–3.623, *p* = 0.0059, Figure 2K) vs (N1 + NX: HR = 2.138, 95% CI: 1.131–4.042, *p* = 0.006, Figure 2L).

### 3.3. The Independent Prognostic Value of ANO4 Expression Evaluated by Univariate and Multivariate Cox Logistic Regression Analysis

Univariate Cox logistic regression analysis of the OS identified five independent prognostic indicators, including age (HR: 0.418, 95% CI: 0.225–0.686, *p* = 0.001), grade (HR: 0.559, 95% CI: 0.378–0.827, *p* = 0.004), stage (HR: 0.388, 95% CI: 0.266–0.566, *p* < 0.001), pT (HR: 0.386, 95% CI: 0.265–0.564, *p* < 0.001), and *ANO4* expression (HR: 2.099, 95% CI: 1.434–3.073, *p* < 0.001), while the gender (*p* = 0.596) and pN (*p* = 0.349) were not prognostic. However, multivariate Cox logistic regression highlighted the age (HR: 0.510, 95% CI: 0.305–0.852, *p* = 0.010) and *ANO4* expression (HR: 1.686, 95% CI: 1.120–2.540, *p* = 0.012) as significant predictors, while the grade (*p* = 0.082), stage (*p* = 0.617), and pT (*p* = 0.912) were not significant. In regard to PFI, univariate Cox logistic regression analysis illustrated a prognostic utility of all indicators except for the pN (*p* = 0.617) which moves in concordance with the OS results. The significant PFI predictors are age (HR: 0.588, 95% CI: 0.360–0.960, *p* = 0.034), gender (HR: 0.621, 95% CI: 0.388–0.992, *p* = 0.046), grade (HR: 0.444, 95% CI: 0.284–0.693, *p* < 0.001), stage (HR: 0.249, 95% CI: 0.163–0.379, *p* < 0.001), pT (HR: 0.264, 95% CI: 0.174–0.402, *p* < 0.001), and *ANO4* expression (HR: 2.521, 95% CI: 1.654–3.840, *p* < 0.001). Upon pooling proposed predictors in PFI multivariate Cox logistic regression, stage (HR: 0.129, 95% CI: 0.092–0.579, *p* = 0.008) and *ANO4* expression (HR: 1.727, 95% CI: 1.103–2.704, *p* = 0.017) were the only significant predictors. A total of six predictors were identified in the univariate Cox logistic regression analysis of the DSS. They include age (HR: 0.097, 95% CI: 0.248–0.995, *p* = 0.048), grade (HR: 0.328, 95% CI: 0.174–0.619, *p* = 0.001), stage (HR: 0.201, 95% CI: 0.112–0.359, *p* < 0.001), pT (HR: 0.210, 95% CI: 0.118–0.374, *p* < 0.001), and *ANO4* expression (HR: 3.894, 95% CI: 2.201–6.889, *p* < 0.001). On the other hand, the DSS multivariate Cox logistic regression portrayed *ANO4* expression as the only predictor (HR: 2.688, 95% CI: 1.465–4.934, *p* = 0.001). Tumor stage was the only predictor in the DFI-related univariate (HR: 0.311, 95% CI: 0.112–0.865, *p* = 0.025) and multivariate (HR: 0.258, 95% CI: 0.079–0.848, *p* = 0.026) Cox logistic regression analysis. Table 3 summarizes the Cox regression results discussed in this section.

### 3.4. ANO4-Related Signaling Pathways Based on GSEA

The first GSEA involved the hallmark gene sets in which a total of 34 out of 50 gene sets were enriched in the low *ANO4* expression group, with 21 gene sets having FDR < 25%. In addition, only 14 gene sets reached the level of statistical significance (*p* ≤ 0.05). The enrichment results include the following sets: epithelial–mesenchymal transition (ES: 0.53, *p* < 0.001, FDR: < 0.001), G2-M checkpoint (ES: 0.47, *p* < 0.001, FDR < 0.001), estrogen response-late (ES: 0.44, *p* < 0.001, FDR = 0.003), E2F targets (ES: 0.45, *p* < 0.001, FDR = 0.003), apical junction (ES: 0.40, *p* < 0.001, FDR = 0.023), glycolysis (ES: 0.40, *p* = 0.001, FDR = 0.023), hypoxia (ES: 0.39, *p* = 0.001, FDR = 0.021), coagulation (ES: 0.40, *p* = 0.004, FDR = 0.024), KRAS signaling pathway (ES: 0.38, *p* = 0.003, FDR = 0.035), complement (ES: 0.37, *p* = 0.005, FDR = 0.044), p53 pathway (ES: 0.37, *p* = 0.007, FDR = 0.042), estrogen response-early (ES: 0.36, *p* = 0.008, FDR < 0.044), myogenesis (ES: 0.36, *p* = 0.008, FDR = 0.054), and TNF-*α* signaling pathway via NF-*κ*B (ES: 0.35, *p* = 0.009, FDR = 0.061). The high *ANO4* expression group enrichment analysis revealed a total of 16 enriched gene sets out of 50, with only three genes with FDR < 25% and a *p* ≤ 0.05. The enriched sets include fatty acid metabolism (ES: 0.39, *p* < 0.001, FDR = 0.043), pancreatic beta cells (ES: 0.48, *p* = 0.014, FDR = 0.024), and bile acid metabolism (ES: 0.39, *p* = 0.004, FDR = 0.037). Figure 3 displays the statistically significant enrichment plots in both the low and high *ANO4* expression groups.

Figure 4A represents the top 15 KEGG-enriched pathways within low and high *ANO4* expression groups. The poor prognostic effect of *ANO4* downregulation is hypothesized to be mediated through the following pathways: ribosome (ES: 0.57, *p* < 0.001), *α*-linolenic acid metabolism (ES: 0.71, *p* = 0.00177), glycosaminoglycan biosynthesis–keratan sulfate (ES: 0.68, *p* = 0.010695), glycosaminoglycan biosynthesis–chondroitin sulfate (ES: 0.61, *p* = 0.006689), complement and coagulation cascades (ES: 0.49, *p* < 0.001), arachidonic acid metabolism (ES: 0.48, *p* = 0.018377), NOD-like receptor signaling pathway (ES: 0.47, *p* = 0.010671), proteasome (ES: 0.49, *p* = 0.024961), basal cell carcinoma (ES: 0.46, *p* = 0.01506), base excision repair (ES: 0.50, *p* = 0.024958), cell cycle (ES: 0.41, *p* = 0.004104), homologous recombination (ES: 0.54, *p* = 0.033333), p53 pathway (ES: 0.44, *p* = 0.023495), taste transduction (ES: 0.46, *p* = 0.018576), and Wnt signaling pathway (ES: 0.38, *p* = 0.007022). On the other hand, the high *ANO4* expression group was associated with significant enrichment of 11 pathways out of the top 15 pathways, with an FDR < 0.25. The pathways are propanoate metabolism (ES: −0.65, *p* < 0.001), valine, leucine, and isoleucine degradation (ES: −0.60, *p* < 0.001), PPAR signaling pathway (ES: −0.54, *p* < 0.001), fatty acid metabolism (ES: −0.54, *p* = 0.00489), peroxisome (ES: 0.56, *p* = 0.005277), renin–angiotensin system pathway (ES: −0.55, *p* = 0.020725), pyruvate metabolism (ES: −0.54, *p* = 0.020725), proximal tubule bicarbonate reclamation (ES: −0.50, *p* = 0.028369), TCA cycle (ES: −0.41, *p* = 0.008772), nitrogen metabolism (ES: −0.48, *p* = 0.01355), and histidine metabolism (ES: −0.54, *p* = 0.026667). Figure 4B represents the GO results highlighting the top five enriched pathways in each subcategory, including MF, CC, and BP.

### 3.5. The PPI Network of ANO4 Based on String Analysis

Figure 5A represents the *ANO4*-related PPI. The generated PPI contains 11 nodes and 12 edges with an average node degree of 2.18, an average local clustering coefficient of 0.883, and PPI enrichment *p*-value of 0.329. The nodes include the following PLSCR5, CCDC181, ACOT7, SLC17A8, ASCL1, ASH2L, TSPAN16, DYSF, SPIC, and GAS2L3.

### 3.6. The Correlation between ANO4 Expression and Tumor Immune Infiltrate

Pre-calculated TCGA immune infiltrate estimates were obtained from xCell. *ANO4* expression correlates significantly with monocyte (*ρ* = −0.1429, *p* = 0.0033) and mast cell (*ρ* = 0.1598, *p* = 0.001) infiltration. On the other hand, non-significant correlations were observed regarding neutrophils (*ρ* = −0.04231, *p* = 0.3859), macrophages (*ρ* = −0.0744, *p* = 0.1270), NK cells (*ρ* = 0.07838, *p* = 0.1079), basophils (*ρ* = −0.09535, *p* = 0.0503), B-cells (*ρ* = −0.02063, *p* = 0.6726), plasma cells (*ρ* = 0.06586, *p* = 0.1769), CD4+ T-cells (*ρ* = 0.02585, *p* = 0.5964), and CD8+ T-cells (*ρ* = 0.04374, *p* =.3700). Figure 4B–K represents the correlation graphs of the tumor microenvironment.

## 4. Discussion

Anoctamins (ANO) are a group of anion channels with eight transmembrane domains, and numerous cellular functions [20]. The coding gene family, formerly known as transmembrane proteins with 16 domains (TMEM16), was only just discovered in 2014 thanks to bioinformatic analysis [31]. The new name (anoctamins), proposed by Yang et al., has replaced TMEM16 in GenBank despite objections, and it has been given the HUGO nomenclature seal of approval [31,32]. This family of transmembrane proteins constitutes ten paralogues (ANO1 through ANO10/ TMEM16A to TMEM16K [excluding the J alphabet from the naming]) dispersed across various human tissues [20]. All anoctamins feature eight hydrophobic helices, which have been previously hypothesized to be transmembrane domains according to hydropathy analysis, however, these findings are still debatable [31]. Duran et al. reported that, unlike ANO1 and ANO2 which have a clear Ca^+2^ activated C^-l^ channel (CaCC) functionality, other members did not pursue such function, since they could not produce a C^-l^ current through Ca^+2^ activation [33]. Furthermore, ANOs 3 through 7 were determined to be intracellular proteins probably residing in the endoplasmic reticulum without being trafficked to the cell membrane [33]. Despite being scarce, this family has been the subject of an expanding body of literature due to its association with numerous pathologies. To illustrate, ANO1 has been linked to a variety of malignancies. Along with ANO5, which has been connected to specific types of muscular dystrophy, ANO6 and ANO10 have also been linked to Scott syndrome and autosomal recessive spinocerebellar ataxia, respectively [31].

Anoctamins have been involved in a wide variety of cellular functions, including epithelial cell secretion, neuronal activation, smooth muscle contractions, skeletal muscle membrane repair, sensory transduction, and carcinogenesis [20,31,33]. On the molecular level, they behave as CaCCs, along with a scramblase activity, in which they express phospholipids across the membrane from the cytoplasmic side to the extracellular side while requiring Ca^+2^ [20,31,33]. *ANO1* has been the most extensively studied ANO protein and has been shown to mediate Cl^−1^ secretion in secretory epithelia of the respiratory, gastrointestinal and renal systems, and sweat glands [34]. Under the activation of noxious heat, its role in heat sensation via somatosensory neurons is obvious [35]. Furthermore, *ANO1* regulates vascular and bronchial smooth muscle tone. Because of this, *ANO1* may play a role in the pathophysiological mechanisms underlying hypertension and asthma, respectively [34,36]. According to Sun *et al*., the interaction between *RANK* and *ANO1* in osteoclasts causes increased bone resorption and decreased bone mass. In individuals with osteoporosis, this makes *ANO1* a viable therapeutic target [37]. *ANO1* antagonist can have a synergistic effect with commercially used tocolytics on human uterine smooth muscles [38]. Finally, and perhaps most importantly, it is believed to escalate multiple carcinogenic processes including cellular proliferation, migration, and metastasis [39]. *ANO2*, a member of the ANO family that is expressed in olfactory sensory cells of the olfactory epithelium may have a function in the olfaction process [40]. It has been found that *ANO3* is expressed in the dorsal root ganglia, which controls nociception [34]. *ANO5* is most abundantly expressed in the musculoskeletal system including bones, chondrocytes, cardiac, and skeletal muscles [41]. The role of *ANO6* as a procoagulant has been confirmed through human and animal studies resulting in Cl^−1^ influx and swelling of platelets [42]. *ANO6* has a well-known scramblase activity, which is also shared with *ANO3*, ANO4, and ANO7 [43]. Overall, except for *ANO1*, which has been thoroughly investigated, other members of the ANO family need even more in-depth research, and we are only at the beginning.

*ANO1*, which is found on locus 11q13, is amplified in multiple tumors, particularly in gastrointestinal stromal tumors (GIST) and head and neck squamous cell carcinoma (HNSCC), in addition to numerous others, such as lung adenocarcinoma, chondroblastoma, esophageal squamous cell carcinoma, salivary gland tumors, oral squamous cell carcinoma, leiomyosarcoma of the uterus, glioma, breast, colorectal and prostate cancer [39,44,45,46,47,48,49,50,51,52,53,54,55,56,57,58,59,60,61]. *ANO1* was a cancer biomarker prior to it is identification as a chloride channel in GIST known as DOG1, and it has received several other names including ORAOV2, and TAOS-2 among oncologists [31,39]. However, its overexpression has, unfortunately, been a poor prognostic factor, leading to greater mortality rates, tumor growth, and invasiveness, as well as its positive correlation with distant metastasis, migration, and tumor grading [45,46,47,50,61]. The carcinogenesis orchestrated by *ANO1* has implicated numerous cell signaling pathways. *ANO1*-mediated proliferation was shown by Duvvuri et al. to be accompanied by extracellular signal-regulated kinase (ERK)1/2 activation, cyclin D1 evocation, and mitogen-activated protein kinase (MAPK) activation [45]. *ANO1* upregulates calcium/calmodulin-dependent protein kinase II (CaMKII) and epidermal growth factor receptor (EGFR) expression, the latter of which regulates the MAPK or PI_3_K-AKT pathway. Therefore, *ANO1* plays a role in governing cell variability through EGFR-AKT/SRC/MAPK and CaMKII signaling pathways [34]. ezrin-radixin-moesin proteins form a cross-link between the cell membrane and actin filaments of the cytoskeleton, hence its involvement in cell migration. Together with its physical association with *ANO1*, this might provide a clue to the role of *ANO1* in EGF-driven migratory and invasive properties [34]. In addition to its connection to the cytoskeleton, its ability to control cell size aids in the development of shrunken cells that can move via diapedesis through inter-endothelial gaps [39,62]. Another intriguing association lies between the sonic hedgehog signaling pathway and *ANO1* which is known to coordinate cellular growth and differentiation [39,63].

Anaplastic thyroid carcinoma appears to acquire its undesirable traits and aggressiveness through *ANO1* overexpression; *ANO1* knockdown greatly reduces the tumor’s aggressive behavior [64]. In gastric cancer, *ANO5* has been linked to a negative prognostic role; when it is knocked down, it causes apoptosis, reduces cell proliferation, and arrests the cell cycle at the G1/S transition [65]. According to research by Pan *et al*., osteosarcomas express *ANO5*, a pro-tumorigenic factor that increases tumor size, grade, and metastasis. The instability and destruction of nel-like proteins 1 and 2 enable such activity [66]. *ANO5* expression is scarce in healthy pancreatic tissue, but it is elevated in pancreatic cancer, where it contributes to the disease’s proliferation and migration [67]. *ANO7* is only expressed in prostate cells, has an unknown function, and is negatively correlated with the prognosis of prostate cancer. Its low expression has been linked to positive surgical margin, lymph node metastasis, high classical and quantitative Gleason grades, advanced tumor stage, high Ki67 labelling index, and early biochemical recurrence [68].

We focused our bioinformatic research on *ANO4* because it is one of the ANO family members that has not received enough attention. It is mostly expressed in the cervix, ovaries, prostate, adrenal glands, and central nervous system [69]. Given this, it has been suggested that it contributes to a variety of neurological diseases, such as schizophrenia, multiple sclerosis, Alzheimer’s disease, and anxiety disorders [70,71,72,73,74]. Reichhart et al. reported that *ANO4* acts as a Ca^2+^-dependent phospholipid scramblase and monovalent nonselective ion channel [69]. Leitzke et al. reported *ANO4* to modulate disintegrin-like metalloproteases ADAM 10 and 17 sheddase activity, evident through the increasing ADAM10 and 17 substrates: transforming growth factor alpha (TGF-*α*), amphiregulin (AREG), and betacellulin. They also demonstrated that the effects brought upon by the overexpression of *ANO4* are due to it is scramblase activity, consequently resulting in diminishing *AREG* and increased cellular proliferation [75]. Maniero et al. recognized *ANO4* as a significant gene expressed in the zona glomerulosa cells despite its contradictory effects on aldosterone secretion [76,77]. To the best of our knowledge, this is the first article to relate the expression of *ANO4* with tumor involvement.

In conclusion, *ANO4* expression was upregulated in tumor samples compared to normal kidney tissue. Albeit the latter finding, low *ANO4* expression is associated with advanced clinicopathological variables such as tumor grade, stage, and pT. In addition, low *ANO4* expression is linked to shorter OS, PFI, and DSS. Multivariate Cox logistic regression analysis identified *ANO4* expression as an independent prognostic variable in OS (HR: 1.686, 95% CI: 1.120–2.540, *p* = 0.012), PFI (HR: 1.727, 95% CI: 1.103–2.704, *p* = 0.017), and DSS (HR: 2.688, 95% CI: 1.465–4.934, *p* = 0.001). GSEA identified the following pathways to be enriched within the low *ANO4* expression group: epithelial–mesenchymal transition, G2-M checkpoint, E2F targets, estrogen response, apical junction, glycolysis, hypoxia, coagulation, KRAS, complement, p53, myogenesis, and TNF-*α* signaling via NF-*κ*B pathways. *ANO4* expression correlates significantly with monocyte (*ρ* = −0.1429, *p* = 0.0033) and mast cell (*ρ* = 0.1598, *p* = 0.001) infiltration.

## Figures and Tables

**Figure 1 jpm-13-00295-f001:**
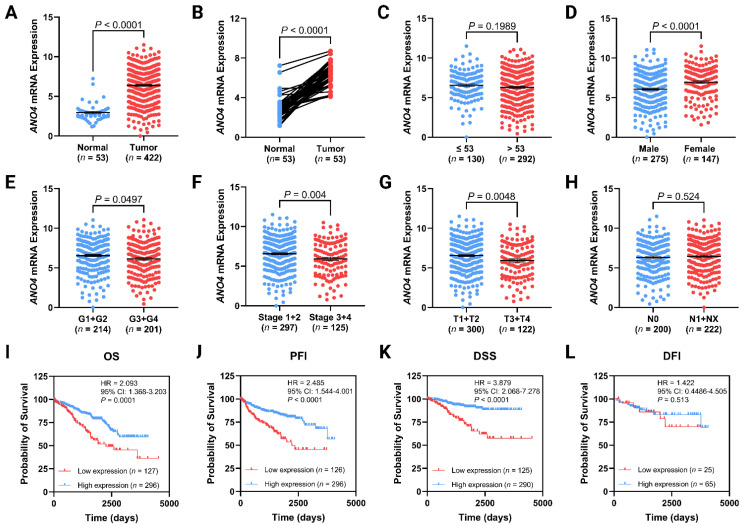
Differential expression of *ANO4* between normal kidney tissues and tumor samples (**A**) and its age and gender-matched comparison (**B**). The correlation between *ANO4* expression and clinicopathological variables including age (**C**), gender (**D**), grade (**E**), stage (**F**), pT (**G**), and pN (**H**). Kaplan–Meier curves demonstrating the impact of *ANO4* expression on the OS (**I**), PFI (**J**), DSS (**K**), and DFI (**L**). Data are presented as mean ± SEM.

**Figure 2 jpm-13-00295-f002:**
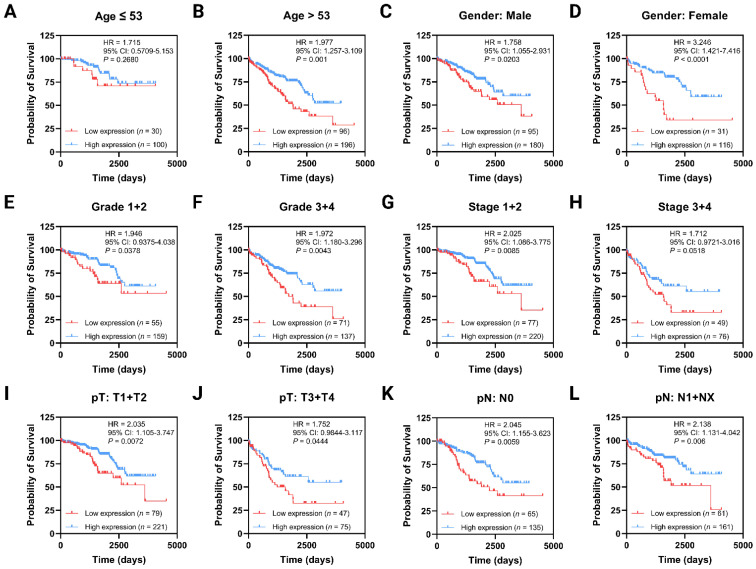
Kaplan–Meier curves demonstrating the impact of *ANO4* expression on the OS within subgroups of dichotomized demographics and clinicopathological variables including age (**A**,**B**), gender (**C**,**D**), grade (**E**,**F**), stage (**G**,**H**), pT (**I**,**J**), and pN (**K**,**L**).

**Figure 3 jpm-13-00295-f003:**
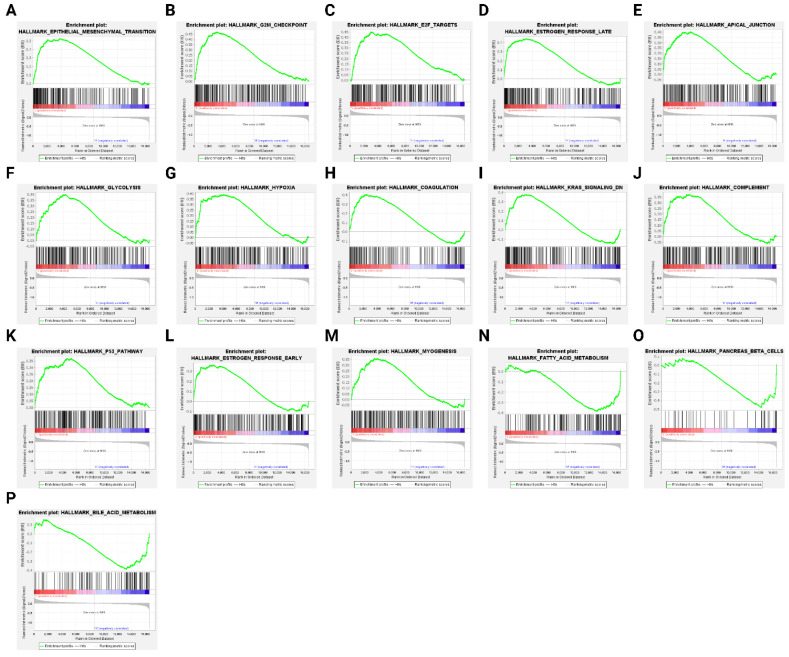
GSEA plots illustrating significantly enriched pathways in low (**A**–**M**) and high (**N**–**P**) *ANO4* expression groups.

**Figure 4 jpm-13-00295-f004:**
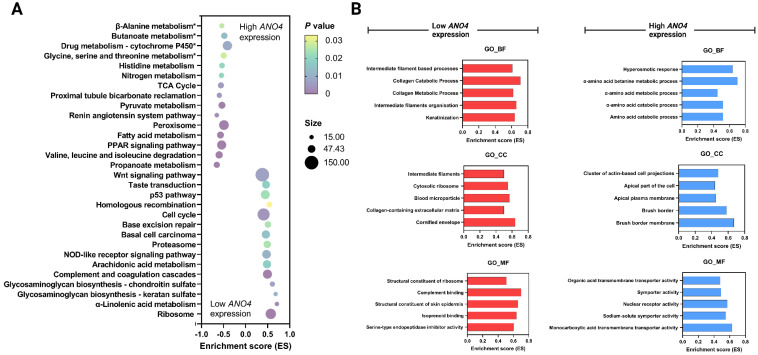
Bubble plot illustrating the top 15 KEGG pathways enriched within the low and high *ANO4* expression groups (**A**). Top five biological processes (BP), cellular components (CC), and molecular functions (MF) enriched within the low and high *ANO4* expression groups (**B**). * Denotes for an FDR > 25%.

**Figure 5 jpm-13-00295-f005:**
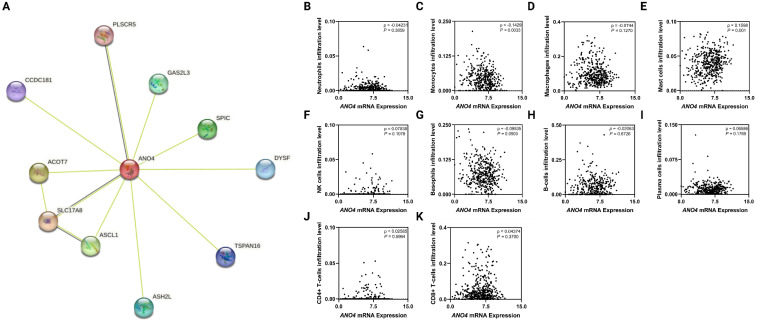
PPI network of *ANO4* based on string analysis (**A**). The correlation between *ANO4* expression and immune microenvironment (**B**–**K**).

**Table 1 jpm-13-00295-t001:** Baseline clinicopathological characteristics of the TCGA non-metastasized ccRCC cohort (*n* = 422).

Characteristic	Data
Age	61 (51–71)
Gender	
Male	275 (65.2)
Female	147 (34.8)
Stage	
1	243 (57.6)
2	54 (12.8)
3	121 (28.7)
4	4 (0.9)
Grade	
1	10 (2.4)
2	204 (48.3)
3	162 (38.4)
4	39 (9.2)
Unknown	7 (1.7)
pT	
T1	245 (58.1)
T2	55 (13.0)
T3	119 (28.3)
T4	3 (0.7)
pN	
N0	200 (47.4)
N1	10 (2.4)
NX	212 (50.2)

Data are presented as *n* (%) or median (IQR).

**Table 2 jpm-13-00295-t002:** Baseline clinicopathological characteristics of the TCGA non-metastasized ccRCC cohort according to *ANO4* expression.

Characteristics	Low Expression (*n* = 126)	High Expression (*n* = 296)	χ^2^	*p*-Value
Age				
≤ 53	30 (7.1)	100 (23.7)	4.125	0.042
> 53	96 (22.7)	196 (46.4)		
Gender				
Male	95 (22.5)	180 (42.7)	8.283	0.004
Female	31 (7.3)	116 (27.5)		
Grade				
G1 + G2	55 (13.3)	159 (38.3)	3.682	0.055
G3 + G4	69 (16.6)	132 (31.8)		
Stage				
1 + 2	77 (18.2)	220 (52.1)	7.402	0.007
3 + 4	49 (11.6)	76 (18.0)		
pT stage				
T1 + T2	79 (18.7)	221 (52.4)	6.155	0.013
T3 + T4	47 (11.1)	75 (17.8)		
pN stage				
N0	65 (15.4)	135 (32.0)	1.267	0.260
N1 + NX	61 (14.5)	161 (38.2)		

Data are presented as *n* (%).

**Table 3 jpm-13-00295-t003:** Univariate and multivariate Cox logistic regression analysis evaluating the utility of *ANO4* expression and relevant clinicopathological covariates in predicting the overall survival.

Covariates *	Univariate Cox Logistic Regression			Multivariate Cox Logistic Regression		
	HR	95% CI	*p*-Value	HR	95% CI	*p*-Value
**OS**						
Age (≤53 vs. >53)	0.418	0.225–0.686	0.001	0.510	0.305–0.852	0.010
Gender (Female vs. Male)	1.111	0.754–1.636	0.596	1.116	0.738–1.688	0.602
Grade (G1 + G2 vs. G3 + G4)	0.559	0.378–0.827	0.004	0.689	0.453–1.048	0.082
Stage (1 + 2 vs. 3 + 4)	0.388	0.266–0.566	<0.001	0.596	0.078–4.529	0.617
pT (T1 + T2 vs. T3 + T4)	0.386	0.265–0.564	<0.001	0.891	0.116–6.870	0.912
pN (N0 vs. N1 + NX)	1.198	0.821–1.749	0.349	1.090	0.738–1.610	0.665
*ANO4* expression (Low vs. High)	2.099	1.434–3.073	<0.001	1.686	1.120–2.540	0.012
**PFI**						
Age (≤53 vs. >53)	0.588	0.360–0.960	0.034	0.655	0.388–1.107	0.114
Gender (Female vs. Male)	0.621	0.388–0.992	0.046	0.627	0.381–1.030	0.065
Grade (G1 + G2 vs. G3 + G4)	0.444	0.284–0.693	<0.001	0.628	0.392–1.007	0.054
Stage (1 + 2 vs. 3 + 4)	0.249	0.163–0.379	<0.001	0.129	0.092–0.579	0.008
pT (T1 + T2 vs. T3 + T4)	0.264	0.174–0.402	<0.001	2.472	0.545–11.215	0.214
pN (N0 vs. N1 + NX)	1.112	0.733–1.687	0.617	1.117	0.727–1.718	0.613
*ANO4* expression (Low vs. High)	2.521	1.654–3.840	<0.001	1.727	1.103–2.704	0.017
**DSS**						
Age (≤53 vs. >53)	0.097	0.248–0.995	0.048	0.681	0.325–1.423	0.307
Gender (Female vs. Male)	0.729	0.392–1.357	0.319	0.839	0.436–1.615	0.599
Grade (G1 + G2 vs. G3 + G4)	0.328	0.174–0.619	0.001	0.519	0.260–1.035	0.063
Stage (1 + 2 vs. 3 + 4)	0.201	0.112–0.359	<0.001	0.217	0.026–1.827	0.160
pT (T1 + T2 vs. T3 + T4)	0.210	0.118–0.374	<0.001	1.415	0.166–12.101	0.751
pN (N0 vs. N1 + NX)	1.297	0.738–2.278	0.366	1.146	0.639–2.005	0.648
*ANO4* expression (Low vs. High)	3.894	2.201–6.889	<0.001	2.688	1.465–4.934	0.001
**DFI**						
Age (≤53 vs. >53)	1.494	0.534–4.177	0.444	1.933	0.619–6.024	0.257
Gender (Female vs. Male)	0.485	0.154–1.526	0.216	0.650	0.183–2.307	0.505
Grade (G1 + G2 vs. G3 + G4)	0.552	0.182–1.674	0.293	0.984	0.277–3.496	0.980
Stage (1 + 2 vs. 3 + 4)	0.311	0.112–0.865	0.025	0.258	0.079–0.848	0.026
pT (T1 + T2 vs. T3 + T4)	0.311	0.112–0.865	0.025	-	–	-
pN (N0 vs. N1 + NX)	0.768	0.271–2.177	0.620	1.102	0.346–3.506	0.869
*ANO4* expression (Low vs. High)	1.438	0.482–4.293	0.515	1.543	0.471–5.059	0.474

* Initial sub-covariate was used a reference.

## Data Availability

Not applicable.

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
