# Peer review of "ANO4 Expression Is a Potential Prognostic Biomarker in Non-Metastasized Clear Cell Renal Cell Carcinoma"

_jpm, 2023, doi:10.3390/jpm13020295_

Round 1

Reviewer 1 Report

The references (2, 3, 4, 12, 13, 14) need to be updated for new therapeutic options.

I would suggest to edit:

- line 5 on page 2, I would mention RCC with sarcomatoid differentiation that is an aggressive form of RCC with poor prognosis.

- Line 20 on page 2: chemoterapy and radiotherapy are not treatment options.

The description of ANO coding gene family in the discussion is too long, I would focus mostly on ANO4.

Are there any clinical trials on ANO4? Is ANO4 potentially able to lead our everydays clinical decisions?

Author Response

Reviewer #1:

Comment #1: The references (2, 3, 4, 12, 13, 14) need to be updated for new therapeutic options.
Response: The comment has been taken into consideration and new references were incorporated.

Comment #2: line 5 on page 2, I would mention RCC with sarcomatoid differentiation that is an aggressive form of RCC with poor prognosis.
Response: RCC with sarcomatoid were mentioned within the introduction part as requested.

Comment #3: Line 20 on page 2: chemotherapy and radiotherapy are not treatment options.
Response: The comment has been taken into consideration and the sentence was modified.

Comment #4: The description of ANO coding gene family in the discussion is too long, I would focus mostly on ANO4.
Response: Unfortunately, the ANO family protein is poorly studied within the literature and we tried to incorporate all the available resources. All the literature about ANO4 were added and discussed thoroughly.

Comment #5: Are there any clinical trials on ANO4? Is ANO4 potentially able to lead our every day’s clinical decisions?
Response: No clinical trials truly evaluated the effect of ANO4 in the prognosis and progression of any malignancy till now. We hope that our results will be a nidus to incorporate ANO4 expression in patient’s prognosis assessment as a mono-variable or added to many gene-based models.

Reviewer 2 Report

1. Please tell me where you took the samples of normal tissue for the differential diagnosis of ANO4 expression between normal tissue and tumor tissue? Is it about conditionally normal kidney tissue that surrounds the tumor?

2. How functionally and phenotypically is ANO4 different from other members of this family?

3. Since we are talking about clear cell renal carcinoma, have you compared the expression of ANO4 with the expression of receptors of the epidermal growth factor family - as an important marker of epithelial neoplasms

4. Did you study the phenotype of infiltrated macrophages (M1 or M2) and compare them with the level of ANO4 expression

5. In the title of the article, the authors announce that ANO4 Expression is a Potential Prognostic Biomarker in Nonmetastasized Clear Cell Renal Cell Carcinoma, and in the results and discussion they only point out that this is one of the first works where ANO4 expression was studied and compared to others markers of clear cell kidney cancer. So is ANO4 a prognostic marker or not?

6. It is desirable to discuss the obtained results in more detail
